# Knowledge heterogeneity and corporate innovation performance: The mediating influence of task conflict and relationship conflict

Rui Gan[1], Xiaoheng Chen[2], Zhiyan Wang [ID][3], Xing Zhang [ID][4,5] *

1 GuiZhou Light Industry Technical College, Guiyang, China, 2 School of Finance and Economics, Guangdong Polytechnic Normal University, Guangzhou, China, 3 Institute of Mining Engineering, Guizhou Institute of Technology, Guiyang, China, 4 School of Management, Guizhou University of Commerce, Guiyang, China, 5 Faculty of Business, City University of Macau, Macau, China

* 201710616@gzcc.edu.cn

**Data Availability Statement:** All relevant data are within the paper.

**Funding:** This study was supported by High-level Talent Project of Guizhou Light Industry Technical

## Abstract

Innovation has emerged as a crucial factor in the sustenance and growth of enterprises. Nonetheless, small and medium-sized enterprises (SMEs) confront numerous challenges in their pursuit of innovation, owing to constraints in capital, expertise, and knowledge resources. Drawing on the resource-based theory and the input-process-output (IPO) model, this study devises a mechanism model to assess the impact of knowledge heterogeneity and innovation performance on small and medium-sized manufacturing enterprises in Guizhou Province, China. The objective is to offer recommendations for the advancement and innovation of enterprises with relative knowledge resource deficiencies. A total of 324 valid questionnaires were gathered, and the acquired data were analyzed employing SPSS 23.0 and Amos 26.0. The findings reveal that knowledge heterogeneity exerts a significantly positive influence on innovation performance. Task conflict and relationship conflict serve as partial mediators in the effects of knowledge heterogeneity on innovation performance. By capitalizing on the heterogeneity of internal and external knowledge, enterprises can effectively enhance their innovation outcomes. Furthermore, the study demonstrates that knowledge sharing possesses a moderating effect on the impact of knowledge heterogeneity on task conflict, relationship conflict, and innovation performance. In a conducive sharing environment, the ultimate effect of knowledge heterogeneity on innovation is subject to alteration.

## 1. Introduction

Innovation is increasingly recognized as a vital approach for enterprises to navigate market competition effectively. The transformative effects of innovation can potentially create novel opportunities for enterprises [1]. In fact, innovation is heavily reliant on the enterprise's knowledge base, encompassing its scope, depth, and degree of heterogeneity [2]. Small and

College (Grant No. 23QYGCC01), and Innovation Project (Education and Science Research) of Guangdong Province (Grant No.2018GXJK098).

**Competing interests:** The authors have declared that no competing interests exist.

medium-sized enterprises (SMEs), are recognized as the driving force behind innovation. However, compared to larger organizations, SMEs typically have restricted access to knowledge, which results in insufficient depth and a fragmented composition of knowledge. Consequently, numerous studies have focused on identifying strategies to assist SMEs in overcoming these knowledge barriers and fostering innovation.

The view of the impact of heterogeneous knowledge on SMEs innovation are mainly positive and inverted U-shaped effects. However, most of the related research focuses on the acquisition of knowledge resources, advocating that companies increase knowledge stock to promote innovation. In fact, from a dynamic perspective, the transformation of a company's knowledge resources into innovative results requires a series of processes such as constant innovation and succession. Many literature lacks corresponding discussions on the intrinsic mechanism of this part. Some literature considering intermediary factors mostly focuses on knowledge coordination or absorption application in the field of knowledge management within the organization. The impact of knowledge heterogeneity on the existing order of the organization is still a research gap.

To fill this gap, this research innovatively discusses the impact mechanism of knowledge heterogeneity on corporate innovation performance with task conflict and relationship conflict as intermediary factors. The main consideration is that in the Chinese context, a higher degree of knowledge heterogeneity within the organization will not only cause differences in opinions among employees when dealing with work tasks but also cause conflicts between members in non-working states.

This study examines the growth of small and medium-sized manufacturing enterprises in Guizhou, a western region in China. Two aspects are mainly considered. One is that Guizhou region itself is an area with a high degree of knowledge heterogeneity. The various ethnic groups in this region have significant differences in culture, lifestyle, and ideology. The study of the application of knowledge heterogeneity by enterprises in this region is more typical. At the same time, Guizhou is located in the southwest inland, which is a region relatively poor in knowledge, education, and talent. The development of enterprises in the region is subject to many environmental and resource constraints. Different from other manufacturing provinces in the country, the region is mainly based on small-scale private capital and it is difficult to form a scale economy to gain competitive advantages. This also means that it is more necessary for the enterprises in this region to play the heterogeneity advantages of the region, solve the problems of innovative development through knowledge transformation.

This study will focus on explaining the following issues:

RQ1: How does knowledge heterogeneity correlate with the innovation performance among manufacturing enterprises situated in Guizhou?

RQ2: In what manner does knowledge heterogeneity steer the ultimate innovation performance via the behavior within the organization and its members?

RQ3: Can positive situational factors promote the transformation of knowledge heterogeneity within the organization?

## 2. Literature review

### 2.1 Resource-based view

The traditional resource-based view emphasizes the importance of resources [3]. In recent years, the application of the resource-based view has focused more on the impact of knowledge and technology as core resources on corporate innovation. With the development of theory,

dynamic resource-based view and resource action view have gradually formed. The dynamic resource-based view advocates that enterprises use their existing internal and external specific capabilities to cope with environmental changes [4]. The resource action school emphasizes process-oriented resource behavior, mainly focusing on the dynamic paths of resource acquisition and integration [5]. The most representative theories in the resource action school are the resource bricolage theory proposed by [6] and the resource orchestration theory proposed by [7]. The former emphasizes that enterprises carry out creative integration of existing resources through resource bricolage. The latter advocates the effective use of resources [8].

Influenced by the resource-based view, this study, when exploring the development of innovation performance of SMEs in Guizhou, focuses on the effective use of knowledge resources. Fully considering the actual characteristics of the local resource structure, we explore how to let local enterprises use resource advantages to promote innovation. Specifically, inspired by the resource bricolage theory, the research focuses on exploring the effective reorganization of enterprise resources in the region. At the same time, under the influence of resource orchestration theory and dynamic capability perspective, this study also fully considers the dynamic changes in the transformation and interaction process of knowledge resources in organizations.

## 2.2 The impact of knowledge heterogeneity on corporate innovation performance

Knowledge heterogeneity refers to the varied characteristics of knowledge across different enterprises, attributable to the disparities in their constituent components [9]. Knowledge heterogeneity encompasses the breadth of knowledge, skills, and expertise available to organizational members [10]. Such heterogeneity can expedite the recognition of novel opportunities and engender unique innovation advantages within organizations.

The impact of knowledge heterogeneity on innovation performance has not yet reached a consensus. According to current research, the impact of heterogeneous knowledge on corporate innovation mainly shows two views: positive impact and inverted U-shaped impact. Studies on the positive effects of knowledge heterogeneity mainly focus on the increase of knowledge stock, triggering benign conflicts, and promoting corporate innovation, etc.

Knowledge heterogeneity in the team can often bring richer perspectives, promote the structural complementarity of knowledge [11], broaden organizational cognition under the interactive influence of different views, and help organizations generate more knowledge aggregation [12]. At the same time, knowledge heterogeneity is considered to help enterprises break away from groupthink [13–15], and alleviate the path dependence in innovation [16]. Especially in innovative entrepreneurial organizations, knowledge heterogeneity facilitates members to appraise problems from a broad spectrum of perspectives and levels, thereby aiding organizations in transmuting individual differentiated thinking into innovative impetus and enhancing team creativity [17, 18].

At the same time, some scholars have proposed the inverted U-shaped effect of knowledge heterogeneity on innovation performance [19, 20] (Ye, Ren & Hao, 2015;. Some research suggests that excessive knowledge heterogeneity can impede organization members' ability to effectively absorb new knowledge, leading to knowledge overload,communication difficulties [21] and a stagnation in the internal exchange of resources [22]. Compared with this, external knowledge of medium-low heterogeneity is more likely to bring breakthrough perspectives to enterprises and make enterprises have a better learning atmosphere [23]. But some research believes that, influenced by collectivist values, organization members in the Chinese context usually have a strong sense of belonging to the organization, so they have a higher tolerance

for heterogeneous knowledge [24, 25]. And Guizhou is a diversified region with high heterogeneity characteristics. Long-term exposure to the atmosphere of ethnic integration makes the enterprises in this environment have stronger acceptance and application capabilities for heterogeneous knowledge.

Therefore, this study proposes the following hypothesis:

H1: There is a positive correlation between knowledge heterogeneity and firm innovation performance.

## 2.3 The mediating role of task conflict and relationship conflict

In the communication process within an organization, friction and conflict are inevitable outcomes of interactions among individuals due to differences in cognition, ideas, and thought processes. In studies pertaining to team conflict, conflict is typically viewed as a bridge that connects the situational antecedents and organizational outcomes of individuals or organizations, thereby playing a mediating role. Deep-rooted differences or antagonisms within the organization surface in the form of conflicts, subsequently affecting various enterprise indicators such as organizational innovation [26, 27]. Task conflict refers to the conflict arising from differences in views and judgments about task content among individuals [28]. The emergence of conflicts in organizations is partly due to differences in the perspectives of individuals looking at issues, and the other part is due to differences in the content covered by knowledge itself [29].

Most literature suggests that task conflict has a positive impact on innovation performance. Task conflicts triggered by diversified knowledge from different fields can accelerate the flow of information among organization members through frequent communication and interaction, stimulate the burst of new ideas, and provide more options for the organization [30]. At the same time, it helps members to re-examine their views from different perspectives [31]. Of course, some literature points out that when the knowledge heterogeneity within the organization is too high, it may strengthen the opposition of views among members, thereby intensifying relationship conflicts, disrupting organizational harmony, and negatively affecting the innovative performance of the enterprise.

However, a review of domestic empirical studies found that most scholars still believe that task conflict has a positive impact on corporate innovation performance, rather than an inverted U-shaped effect. This is because, influenced by the doctrine of the mean, most Chinese corporate cultures, including corporate members, advocate harmony [32] and Chinese members care more about collective interests rather than personal interests in the organization. Therefore, organization members usually do not allow conflicts to develop to an irreparable situation. When conflicts escalate, some people tend to avoid and tolerate conflicts, and other members in the organization will also help the organization to alleviate conflicts [33].

Therefore, this study proposes the following hypothesis:

H2: Knowledge heterogeneity positively influences task conflict;

H3: Task conflict positively impacts firm innovation performance;

H4: Task conflict mediates the positive relationship between knowledge heterogeneity and firm innovation performance;

Relationship conflict refers to the emotional opposition arising from differences in values and ideologies among individuals [34]. In the study of the causes of relationship conflict, a common viewpoint is that heterogeneity among individuals is the primary factor leading to relationship conflict [35, 36], with the main manifestation being knowledge differences.

Compared with task conflicts, relationship conflicts are generally considered to be mixed with stronger personal emotions and feelings [37]. Relationship conflict can have a significant negative impact on members' psychological empowerment and team trust, thereby making members reluctant to actively share innovative ideas, inhibiting corporate innovation [38, 39].

At the same time, affected by relationship conflict, members may lack rational thinking when discussing work tasks and resort to subjective emotional judgments. Some studies have also found that a large number of Chinese corporate employees, like introverted members, show an attitude of avoidance and tolerance towards relationship conflict. Even if they have opinions about the other's behavior, as long as it is not completely unbearable, they will choose to give in [40]. However, this tolerance does not mean the disappearance of conflict, but is reflected in the rejection and resistance to the relevant individual [41]. From the perspective of corporate innovation, the mutual exclusion of members will make the organizational atmosphere rigid, cooperation inefficient, and hinder the development of corporate innovation performance. Therefore, when the knowledge heterogeneity in the organization has caused intense relationship conflicts, its original positive effect on corporate innovation performance may be inhibited to a certain extent.

Therefore, this study proposes the following hypothesis:

H5: Knowledge heterogeneity positively affects relationship conflict;

H6: Relationship conflict negatively impacts firm innovation performance;

H7: Relationship conflict mediates the relationship between knowledge heterogeneity and firm innovation performance, albeit negatively;

## 2.4 The moderating role of knowledge sharing

Knowledge sharing denotes the exchange and circulation of knowledge between sender and receiver (among individuals, teams, and organizations), which entails the receiver's continuous processing and integration of knowledge, eventually metamorphosing into skills or cognition required by the organization. Adequate knowledge sharing can efficaciously enhance the efficient transmission of explicit and tacit knowledge, assuage potential conflicts arising from the communication of heterogeneous knowledge, and exert a positive moderating impact on innovation performance [42]. A positive organizational environment is perceived to stimulate team members' sense of belonging and bolster the influence of knowledge heterogeneity on innovation. Tsai [43] observed that an absence of a sharing environment entails that even if members maintain cordial relationships, they may lack the interaction of work-related perspectives. Consequently, even for teams exhibiting high knowledge heterogeneity, if members cannot actively and effectively articulate their ideas, the innovation performance derived from heterogeneity is comparatively constrained [44].

A harmonious sharing environment can foster not only corporate innovation but also effectively influence a series of conflicts within the organization. From the perspective of shared mental models, with frequent sharing and interaction, communication barriers among members can be effectively reduced, mitigating potential relational conflicts due to insufficient information exchange [45]. Bandura's theory of reciprocal determinism also posits that the environment plays a role in influencing individual behavior [46]. In a harmonious and open knowledge-sharing context, organization members are encouraged to promote task-related discussion, actively provide work suggestions, and enhance the effect of task conflict. From the perspective of personal emotion, a conducive sharing atmosphere promotes emotional communication among employees, making it easier for members to resonate, thus reducing the frequency and severity of relational conflicts.

Therefore, this study proposes the following hypothesis:

H8: Knowledge sharing strengthens the positive relationship between knowledge heterogeneity and innovation performance;

H9: Knowledge sharing augments the positive relationship between knowledge heterogeneity and task conflict;

H10: Knowledge sharing diminishes the positive relationship between knowledge heterogeneity and relationship conflict.

## 3. Methods

### 3.1 Model framework

In the study, we employ the Input-Process-Output (IPO) model paradigm, integrating it with the resource-based theory to formulate relevant hypotheses. The IPO model, initially introduced by McGrath in his seminal work "Social Psychology: An Introduction," advocates for investigating the origins of influences on organizational performance through the 'input'–'process'–'output' model paradigm. Building on McGrath's IPO model, West, Hirst, Richter & Shipton [47] centered their research on the pathways through which various organizational elements impact innovation performance (refer to Fig 1). Their work further refined the 'process' section, thereby expanding the empirical analysis of this model in the context of team innovation research.

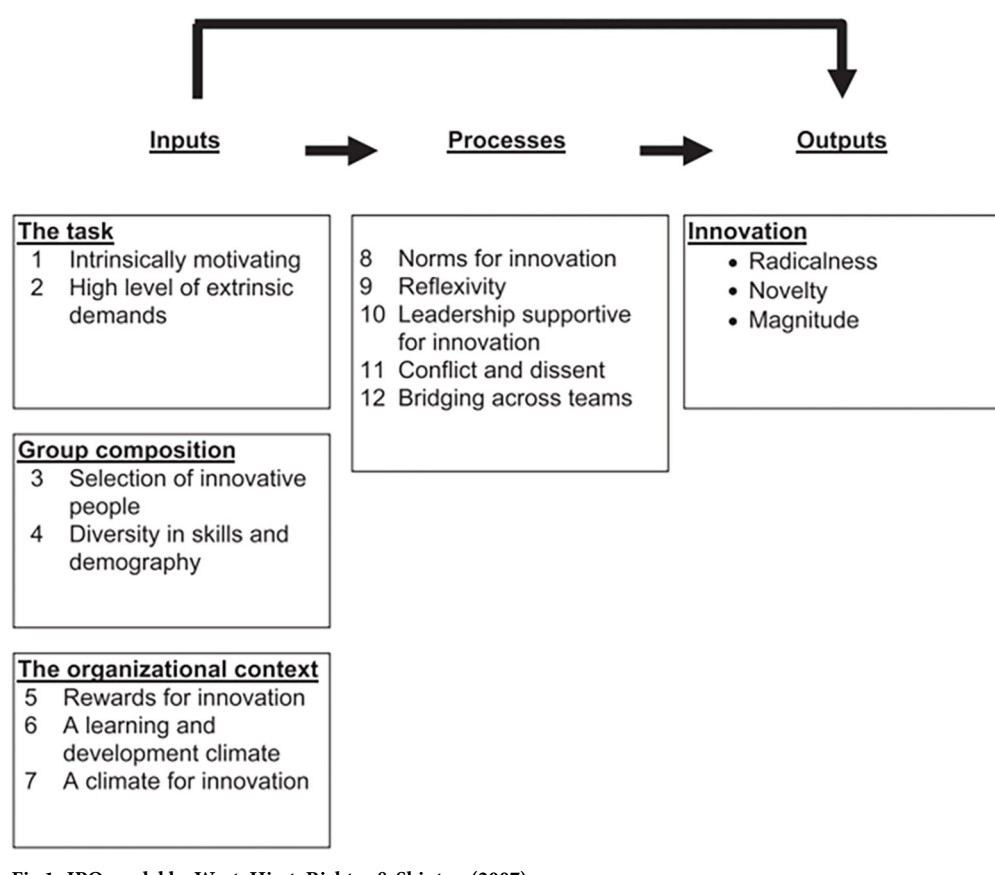

**Fig 1. IPO model by West, Hirst, Richter & Shipton (2007).**

In the framework proposed by West, the 'process' segment primarily encompasses a range of intra-organizational behaviors, viewed as crucial mechanisms for transforming 'input' into 'output.' These include innovation norms, leadership support, reflection, conflict and dissent, and teamwork. Innovation norms emphasize the role of standardized team behavior in enhancing an organization's innovative acumen. Organizational reflection signifies the collective retrospection by the team and its members on goal-setting, strategy execution, and implementation, fostering better self-regulation and improve the organization's proactive and adaptive capacities. Leadership support underscores the necessity for a leadership style that fosters innovation among team members while maintaining adequate oversight and control. The focus of conflict and dissent is on constructive disagreement, suggesting that conflict serves a pivotal role in fostering organizational innovation and advocates for the active expression of diverse viewpoints within the organization. Lastly, teamwork alludes to the team's coordination and mutual assistance. West et al. highlight the need for a balanced team atmosphere. On one end, excessive internal dissent may foster organizational divisions. Conversely, an overwhelming consensus may engender groupthink, detrimental to innovation. Thus, achieving a balance between these two extremes is of paramount importance.

While the traditional resource-based theory underscores the significance of resource elements in enterprise development, the resource action perspective suggests that strategic resource allocation is crucial for leveraging resource advantages. West et al.'s IPO model aptly integrates these perspectives, establishing a systemic mechanism. However, while the model delineates organizational factors potentially impacting innovation performance and their respective processes, it does not extensively explore the precise means through which each factor influences innovation.

To further validate and expand upon the IPO model, this study will scrutinize specific 'input' elements. Given the focus of this research and the particular attributes of small and medium-sized manufacturing firms in Guizhou Province, the study concentrates on facets of population diversity. Drawing from a literature review, we introduce and define the concept of knowledge heterogeneity, positing it as an independent variable affecting innovation performance. Furthermore, the IPO model proposed by West and colleagues emphasizes the positive role of conflict within the 'process' phase, endorsing the beneficial use of constructive conflict to foster innovation. As an extension to this, we categorize conflict into task and relationship conflict. Using the IPO model's logic, we aim to examine how knowledge heterogeneity contributes to innovative outcomes via the process of conflict. Moreover, we propose knowledge sharing as a moderating variable, investigating its moderating impact on the mediator and dependent variables, thus constructing the model framework for this study.

Drawing from the IPO model paradigm, resource-based theory, and a comprehensive literature review, the following hypotheses are postulated in this study (Fig 2.).

## 3.2 Research method

Building upon theoretical research and the input-process-output model, this study formulates a fundamental model framework, posits associated research hypotheses, and identifies the relationships amongst variables including knowledge heterogeneity, innovation performance, task conflict, relationship conflict, and knowledge sharing. Currently, these variables have been subject to extensive investigation within the realm of management. As such, the measurement of each variable can rely on the selection of mature scales available in existing literature.

In line with Brown's [48] recommendation, a seven-point Likert scale was employed in the present investigation, enabling participants to make selections reflecting their enterprises' actual conditions. These scores spanned a range from low to high.

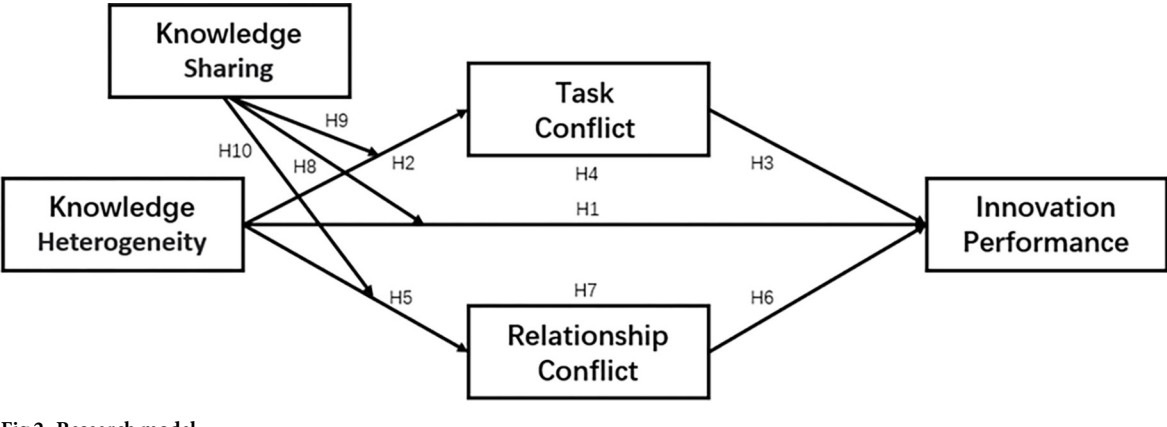

**Fig 2. Research model.**

Innovation performance was quantified employing the scale established by Qian, Yang, and Xu [49]. The creation of this scale was informed by the work of Bell [50] and Ritter & Gemunden [51], and it has demonstrated robust reliability and validity through an exploratory factor test. Ye, Hao, and Patel's [52] comprehensive nine-item scale was deployed to assess knowledge heterogeneity, encompassing internal (four items) and external (five items) facets. Respondents were tasked with completing items relative to their enterprise's internal knowledge heterogeneity over the past three years and external knowledge heterogeneity in relation to their collaborative enterprise. This scale has demonstrated satisfactory reliability and validity. Jehn's [28] seminal scale was employed to measure task and relationship conflict, encompassing four items for each construct. Knowledge sharing was gauged using Faraj's four-item scale, a pioneering systematic measure of this construct. Subsequent work by Faraj & Sproull [53], and Wickramasinghe & Widyaratne [54], has further corroborated its applicability and consistency. Srivastava, Park, and Yun [55] and Lin & Shin [56] have also utilized this scale, either partially or entirely, in their investigations of knowledge sharing.

The survey was conducted across six prefectural cities (Guiyang, Zunyi, Bijie, Anshun, Tongren, and Liupanshui) and three ethnic minority autonomous prefectures (Qiannan Buyei-Miao, Qiandongnan Miao and Dong, and Qiandinan Buyei-Miao) in China's Guizhou Province. A sum of 885 questionnaires was disseminated both online and offline, yielding 457 responses, a recovery rate of 51.64%. After discarding 38 samples with incomplete responses and 76 samples displaying evident contradictions or patterns in their responses, 324 valid questionnaires remained, representing an effective sample rate of 36.61%.

## 4. Results

### 4.1 Reliability test

Reliability, pertaining to the dependability of a questionnaire, is typically reflected in the internal consistency of test outcomes. An effective measure should yield consistent results for repeated measurements, thus providing credibility to the gathered data. Inconsistent results denote weak data stability. Among the various methods to assess scale reliability, this study employs Cronbach's α coefficient to represent the internal consistency of the scale. A higher α value signifies greater consistency among the questionnaire items, indicating superior scale reliability. An α coefficient lower than .6 represents low reliability, necessitating either a questionnaire revision or the elimination of contentious indicators. Conversely, an α coefficient higher than .8 suggests a high degree of stability in questionnaire data.

**Table 1. Reliability test of each variable.**

| Variable/Dimension | Number of Items | Cronbach's α maximum value after each variable deletion item | Cronbach's α |
|---|---|---|---|
| Innovation Performance (IP) | 5 | .877 | .892 |
| Knowledge Heterogeneity (KH) | 9 | .867 | .877 |
| Internal Knowledge Heterogeneity (IKH) | 4 | .896 | .912 |
| External Knowledge Heterogeneity (EKH) | 5 | .888 | .899 |
| Task Conflict (TC) | 4 | .820 | .840 |
| Relationship Conflict (RC) | 4 | .884 | .902 |
| Knowledge Sharing (KS) | 4 | .851 | .877 |

The aforementioned methods and pertinent criteria were utilized to ascertain the reliability of each variable and dimension in the questionnaire. These findings are outlined in Table 1. The data reveals that all constructs and items in this study achieved satisfactory reliability, with values ranging from .840 to .912. Task conflict and relationship conflict possess reliabilities of .840 and .902, respectively, while knowledge heterogeneity has a reliability of .877. Within this, internal knowledge heterogeneity has a reliability of .912, and external knowledge heterogeneity, .899. Further, the values for innovation performance and knowledge sharing exceed .8, suggesting that the questionnaire's measurement stability for each variable is high, thereby conferring credibility. The table also discloses the maximum Cronbach's α value following the deletion of each variable item. As per these results, the α value for each variable, post-item deletion, is lower than the α value for the complete item, further endorsing the validity of the item.

## 4.2 Validity test

Following the demonstration of the measuring tool's reliability, it is imperative to assess its validity. A highly valid measuring instrument concentrates on the true objective of measurement, rather than diverging into other constructs, and exhibits enhanced evaluation capabilities of the target construct. This study appraises both the content validity and construct validity of the questionnaire.

Content validity is assessed to confirm that the literal interpretation of all items accurately portrays the measured constructs. To enhance the questionnaire's content validity, this study utilized a mature scale, one that has been validated and widely adopted in recent scholarly works. Given that the questionnaire includes both English and Chinese scales, systematic translation and iterative comparative discussions were conducted to ensure the preservation of information throughout the language unification process. Before the official questionnaire distribution, feedback obtained from pre-test respondents was collected for refinement, thereby maximizing the questionnaire's content validity.

Construct validity measurement tests the congruence between the gathered data and the theoretical underpinnings. In this study, exploratory factor analysis and confirmatory factor sub-analysis will be initially performed, followed by an evaluation of the model's fit indices. Subsequently, convergent validity and discriminant validity will be examined.

**4.2.1 Confirmatory Factor Analysis (CFA).** Confirmatory factor analysis is employed to examine whether the relationship between a factor and its corresponding observed variable aligns with the researcher's theoretical presuppositions. In this study, confirmatory factor analysis was utilized to compute the convergent and discriminant validity of each latent variable. Generally, a standardized load of the observed variable greater than .5 indicates sound construct validity for the scale. Combined reliability (CR) is another criterion for assessing scale

quality, reflecting the extent to which all observed variables within each latent variable consistently interpret the latent variable. A combined reliability CR greater than .7 suggests that all observed variables within each latent variable can consistently interpret the latent variable [57].

Average variance extracted (AVE) is employed to gauge the convergent and discriminant validity of each latent variable, primarily explaining the proportion of latent variable variance due to measurement error. Generally, the larger the AVE, the smaller the relative measurement error. Past research suggests that if the latent variable's AVE is greater than .5, and the square root of the AVE is larger than the correlation coefficient between the latent variable and other latent variables, the scale exhibits strong convergent and discriminant validity.

In this study, AMOS26.0 software was used to conduct a confirmatory factor analysis of the scale, and a confirmatory factor model was constructed based on the exploratory factor analysis results. The suitability of the confirmatory factor model established in this paper was ascertained by evaluating the fit indices of the structural equation. Meeting the established criteria would signify that the constructed model could effectively measure the pertinent latent variables.

The fitting index of model operation is shown in Table 2. The fitting index is $x^2$/df = 1.429, less than 5, and less than 3, and the effect is good. GFI = .913, AGFI = .892, NFI = .927, TLI = .973, CFI = .977, all greater than .8, RMSEA = .036. Is less than .08, according to the fitting standard, the fitting indexes of the model of confirmatory factor analysis meet the requirements, which is suitable for the model.

**4.2.2 Convergent validity test.** Convergent Validity means that when two different measuring tools are used to measure the same concept, the classification is highly correlated. In this study, convergence validity was tested by measuring combination reliability (CR) and mean variance extraction value (AVE). The combined reliability is usually greater than .7 [57], and AVE is greater than .5 [58], which is the reference value to meet the standard.

The observed factor load, combined reliability (CR) and mean variance extraction (AVE) of all variables in the overall scale are shown in Table 3. Factor loading values of each item range from .674 to .881, indicating high convergent validity, CR of each dimension is greater than .7, AVE is greater than .5, indicating that the scale in this study has good convergent validity.

**4.2.3 Discriminative validity test.** The test of discriminant validity requires a comparison of the AVE value in the correlation matrix between image items with the Pearson correlation. If the average correlation among items within the same variable exceeds the correlation between variables, discriminant validity is established. This signifies that the inter-item correlation within the variable surpasses the correlation between dimensions.

As depicted in Table 4, the square root of the AVE for each dimension exceeds the correlation coefficient between dimensions, indicating strong discriminant validity for the scale.

**4.2.4 Homologous variance test.** Given that all questionnaire items were responded to by the same individual, systematic errors were mitigated to avoid covariation resulting from external environmental interference. This study employed Harman's single factor analysis method to test for common method variance in the data. All questionnaire items were subjected to exploratory factor analysis, resulting in the extraction of six principal component

**Table 2. Model fitting index of the global scale.**

| Index | $x^2$/df | GFI | AGFI | NFI | TLI | CFI | RMSEA |
|---|---|---|---|---|---|---|---|
| Statistic | 1.429 | .913 | .892 | .927 | .973 | .977 | .036 |
| Standard | <5 | >.8 | >.8 | >.8 | >.8 | >.8 | < .08 |
| Result | Fitting | Fitting | Fitting | Fitting | Fitting | Fitting | Fitting |

**Table 3. Test of convergent validity.**

| Variable/Dimension | Items | b | S.E. | C.R. | P | β | CR | AVE |
|---|---|---|---|---|---|---|---|---|
| EKH | EKH1 | 1 | | | | .844 | .900 | .643 |
| | EKH2 | 1.004 | .057 | 17.528 | *** | .822 | | |
| | EKH3 | .872 | .055 | 15.950 | *** | .770 | | |
| | EKH4 | .893 | .059 | 15.216 | *** | .745 | | |
| | EKH5 | .964 | .055 | 17.613 | *** | .824 | | |
| IKH | IKH1 | 1 | | | | .881 | .912 | .722 |
| | IKH2 | .885 | .047 | 18.681 | *** | .813 | | |
| | IKH3 | .914 | .047 | 19.555 | *** | .835 | | |
| | IKH4 | 1.018 | .049 | 20.924 | *** | .868 | | |
| IP | IP1 | 1 | | | | .783 | .893 | .625 |
| | IP2 | 1.014 | .066 | 15.420 | *** | .801 | | |
| | IP3 | .915 | .062 | 15.216 | *** | .792 | | |
| | IP4 | .870 | .062 | 14.130 | *** | .745 | | |
| | IP5 | .998 | .062 | 16.101 | *** | .829 | | |
| KS | KS1 | 1 | | | | .834 | .878 | .642 |
| | KS2 | .979 | .061 | 16.139 | *** | .813 | | |
| | KS3 | .904 | .059 | 15.287 | *** | .778 | | |
| | KS4 | .946 | .062 | 15.303 | *** | .779 | | |
| RC | RC1 | 1 | | | | .852 | .903 | .699 |
| | RC2 | 1.016 | .055 | 18.554 | *** | .843 | | |
| | RC3 | .965 | .056 | 17.354 | *** | .807 | | |
| | RC4 | 1.047 | .056 | 18.541 | *** | .842 | | |
| TC | TC1 | 1 | | | | .833 | .843 | .575 |
| | TC2 | .774 | .062 | 12.485 | *** | .674 | | |
| | TC3 | .905 | .068 | 13.393 | *** | .715 | | |
| | TC4 | .967 | .064 | 15.225 | *** | .800 | | |

Note: * $p < .05$

** $p < .01$

*** $p < .001$.

factors, including two dimensions of independent variables. The variance explanation rate was checked, and the common method variance issue was tested. The results reveal a total explanatory power of the scale of 73.458%, with the contribution rate of the first principal component being 14.215%, which is less than the generally accepted 50%. The contribution rates of the six

**Table 4. Discriminative validity test.**

| | 1 | 2 | 3 | 4 | 5 | 6 |
|---|---|---|---|---|---|---|
| EKH | .802 | | | | | |
| IKH | .380 | .850 | | | | |
| IP | .462 | .467 | .791 | | | |
| TC | .451 | .542 | .540 | .758 | | |
| RC | .101 | .198 | -.502 | -.012 | .836 | |
| KS | .103 | .332 | .304 | .203 | -.016 | .801 |

Note: The diagonal bold is the square root of AVE, and the lower triangle is the Pearson correlation of the variable.

principal components are relatively evenly distributed, thereby ruling out the possibility of common method variance.

## 4.3 Hypothesis testing

In this section, we will test all hypotheses postulated in the study, encompassing main effects, mediating effects, and moderated mediating effects. To maintain a rigorous and scientific approach, control variables were included in these tests. The mediation effect was substantiated through the three-step sequential test by Baron & Kenny [59]. Specifically, the primary effect of the independent variable, knowledge heterogeneity, on the dependent variable, innovation performance, will be first tested. Subsequently, the mediating effect will be evaluated in a stepwise fashion. To ensure the accuracy of the mediating effect data, this study employed regression analysis and bootstrapping for verification. Additionally, to ensure the stability and reliability of the regression analysis results, multicollinearity and serial correlation tests were implemented during the analysis. Tests for primary and mediating effects report Durbin-Watson (dw) values, minimum tolerances, and maximum VIF values for the model. The dw value is utilized to test the sample's independence. A measurement range between 1.5 and 2.5 signifies sample independence, indicating an absence of serial correlation issues. When the minimum tolerance exceeds .5 and the maximum VIF value is less than 10, it is generally considered that the model is free from collinearity problems, which aids in assessing the model's goodness of fit.

**4.3.1 Main effect test.** Two models (M1 and M2) were constructed by regression analysis. The relationship between knowledge heterogeneity and corporate innovation performance is tested, and the relevant results are shown in Table 5. M1 is the basic model, which mainly covers four control variables. The regression results show that the years of enterprise (B = .143, $p<0.01$), employee education (B = .347, $p<0.01$), employee age (B = .160, $p<0.05$) have a significant positive effect on enterprise innovation performance. In M2, knowledge heterogeneity of independent variables is added on the basis of M1. The regression coefficient of

**Table 5. Results of main effect regression analysis (knowledge heterogeneity and innovation performance).**

| Type | Variables | Innovation Performance (Dependent Variable) | | | |
|---|---|---|---|---|---|
| | | M1 | | M2 | |
| | | B | t | B | t |
| Control Variable | Year | .143 | 2.811** | .087 | 1.935 |
| | Education | .347 | 3.080** | .306 | 3.121** |
| | Age | .160 | 2.374* | .195 | 3.315** |
| | R&D | .070 | 1.332 | .061 | 1.341 |
| Independent Variable | KH | | | .511 | 10.065*** |
| $R^2$ | | .086 | | .306 | |
| Adjust $R^2$ | | .074 | | .296 | |
| $\Delta R^2$ | | | | .221 | |
| F | | 7.459*** | | 28.105*** | |
| dw | | 2.220 | | 2.150 | |
| Maximum VIF | | 1.035 | | 1.035 | |
| Minimum Tolerance | | .966 | | .966 | |

Note: * $p < .05$

** $p < .01$

*** $p < .001$.

independent variable to dependent variable of enterprise innovation performance B = .511 and p < .001. It is proved that knowledge heterogeneity has a significant positive effect on firm innovation performance, and hypothesis 1 is valid.

Meanwhile, the dw(Durbin-Watson) values of model are also tested in Table 5, and the results are all within the standard range of 1.5 to 2.5, indicating that the samples in the scale are independent and there is no sequential correlation problem. The minimum tolerance in the model is greater than .5, and the maximum VIF value is less than 5, which proves that there is no collinearity between variables in the regression model and meets the requirements for model fitting.

**4.3.2 Test of mediating effect.** *1. Test of the mediating role of task relationship.* Continuing with the analysis, the sequential method is used to conduct regression tests and examine the mediation effects for knowledge heterogeneity, task conflict, and innovation performance, with the regression results displayed in Table 6. In this table, M1 and M2 refer to regressions with task conflict as the dependent variable, whereas M3 and M4 refer to regressions with innovation performance as the dependent variable.

M1 is the basic model for the mediation variables, including four control variables. The results showed that none of the four control variables had significant effects on task conflict. The knowledge heterogeneity of independent variables was added to M2, and the regression coefficient showed that B = .542, p < .001. That is, knowledge heterogeneity has a significant positive effect on task conflict, and hypothesis 2 is valid. Knowledge heterogeneity can provide multi-dimensional and multi-level viewpoints for organizations. When dealing with work tasks, whether it is the collision of different viewpoints among members or among organizations, it can promote the generation of task conflict.

As in the main effect verification of the previous step, the research has tested the regression analysis of knowledge heterogeneity of control variables and independent variables on innovation performance. Therefore, the results of the main effect will not be repeated in this test. In

**Table 6. Results of regression analysis of mediating effect (knowledge heterogeneity, task conflict and innovation performance).**

| Type | Variable | Task Conflict (Mediator) | | | | Innovation Performance (Dependent variable) | | | |
|---|---|---|---|---|---|---|---|---|---|
| | | M1 | | M2 | | M3 | | M4 | |
| | | B | t | B | t | B | t | B | t |
| Control Variable | Year | .047 | 0.957 | -.013 | -.310 | .121 | 2.665** | .091 | 2.096* |
| | Edu. | .095 | 0.874 | .052 | .572 | .301 | 3.013** | .291 | 3.075** |
| | Age | -.043 | -.667 | -006 | -.111 | .181 | 3.028** | .197 | 3.472** |
| | R&D | .013 | .262 | .004 | .100 | .063 | 1.365 | .060 | 1.363 |
| Independent Variable | KH | | | .542 | 11.444 *** | | | .352 | 6.059 *** |
| Mediator | TC | | | | | .482 | 9.389 *** | .293 | 5.062 *** |
| $R^2$ | | .008 | | .297 | | .284 | | .358 | |
| Adjust $R^2$ | | -.005 | | .286 | | .273 | | .346 | |
| $\Delta R^2$ | | | | .289 | | .198 | | .052 | |
| F | | .613 | | 22.540*** | | 25.231*** | | 29.504*** | |
| Dw | | 2.008 | | 1.950 | | 2.223 | | 2.180 | |
| Maximum VIF | | 1.035 | | 1.035 | | 1.036 | | 1.445 | |
| Minimum Tolerance | | .966 | | .966 | | .966 | | .692 | |

Note: * p < .05

** p < .01

*** p < .001.

M3, task conflict of intermediate variable is added on the basis of control variable, and the regression coefficient of innovation performance is B = .482, p < .001. On the basis of M3, the knowledge heterogeneity of independent variables is introduced again in Model 4, and the regression coefficient of independent variables B = .352, p < .001. Compared with the coefficient B = .542 (p < .001) of M 2 in the table, the value is significantly reduced. The regression coefficient of task conflict of intermediate variable B = .293, p < .001. The coefficients of knowledge heterogeneity of independent variable and task conflict of intermediate variable are both positive. Therefore, the test results prove that task conflict plays a mediating role between knowledge heterogeneity and firm innovation performance, and hypothesis 3 and 4 are valid. Knowledge heterogeneity provides multi-dimensional knowledge for the organization and causes task conflict in the interaction of viewpoints. On the one hand, task conflict can provide rich ideas for innovation and development. On the other hand, it can also help team members to re-examine their own views from different perspectives and improve the quality of decision-making.

Meanwhile, Table 6 presents the Durbin-Watson (dw) values for the model, all of which lie within the accepted range of 1.5 to 2.5. This indicates that the samples in the scale are independent, and there is no autocorrelation issue. The model's minimum tolerance exceeds .5, and the maximum Variance Inflation Factor (VIF) value is less than 5, demonstrating that there is no collinearity between variables in the regression model, satisfying the prerequisites for model fitting. Additionally, since the independent variable, knowledge heterogeneity, in M4 still holds a significant regression coefficient on firm innovation performance, this suggests that task conflict plays a partial mediating role.

*2. Test of the mediating role of relationship conflict.* In this study, the sequential method is used to conduct regression tests and examine the mediation effects for knowledge heterogeneity, relationship conflict, and innovation performance, with the regression results presented in Table 7. Here, M1 and M2 refer to regression analyses with relationship conflict as the dependent variable, while M3 and M4 represent regression analyses with innovation performance as the dependent variable.

M1 is the basic model for the mediation variables, including four control variables. The results showed that employee age had a negative effect on relationship conflict (B = -,604, p < .001). The knowledge heterogeneity of independent variables was added to M2, and the regression coefficient showed that B = .232, p < .001. That is, knowledge heterogeneity has a significant positive effect on relationship conflict, and hypothesis 5 is valid. Knowledge heterogeneity is not only reflected in the differences of viewpoints at work, but also permeates into the daily behavior and interpersonal communication of members. This difference is more likely to induce emotional conflicts within the organization, destroy interpersonal relationships and cause relationship conflicts.

In the main effect verification, the research has tested the regression analysis of knowledge heterogeneity of control variables and independent variables on innovation performance. Therefore, the main effect results will not be repeated in this test. On the basis of control variables, relationship conflict of intermediary variables is added in M3, and the regression coefficient of innovation performance is B = -.381, p < .01. On the basis of M3, the knowledge heterogeneity of independent variables is introduced again in M4, and the regression coefficient of independent variables is B = .625, p < .001. Regression coefficient B = -.491, p < .001. Therefore, the test results prove that relationship conflict plays a mediating role between knowledge heterogeneity and firm innovation performance, and hypothesis 6 and 7 are valid.

In particular, by comparing the regression coefficient of task conflict partially verified above, it is found that the regression coefficient of relationship conflict as an intermediary variable is negative (B = -.491). This suggests that relationship conflict inhibits the positive effect

**Table 7. Results of regression analysis of mediating effect (knowledge heterogeneity, relationship conflict and innovation performance).**

| Type | Variable | Relationship Conflict (Mediator) | | | | Innovation Performance (Dependent variable) | | | |
|---|---|---|---|---|---|---|---|---|---|
| | | M1 | | M2 | | M3 | | M4 | |
| | | B | t | B | t | B | t | B | t |
| Control Variable | Year | -.077 | -1.357 | -.102 | -1.834 | .114 | 2.458* | .036 | 1.021 |
| | Edu. | -.131 | -1.056 | -.150 | -1.224 | .296 | 2.897** | .233 | 2.986** |
| | Age | -.604 | -8.100 *** | -.588 | -8.024 *** | -.070 | -1.049 | -.094 | -1.832 |
| | R&D | -.125 | -2.163* | -.129 | -2.273* | .022 | .458 | .002 | -.062 |
| Independent Variable | KH | | | .232 | 3.678 *** | | | .625 | 15.218 *** |
| Mediator | TC | | | | | -.381 | -8.306 *** | -.491 | -13.759 *** |
| $R^2$ | | .201 | | .234 | | .249 | | .566 | |
| Adjust $R^2$ | | .191 | | .222 | | .237 | | .558 | |
| $\Delta R^2$ | | | | .033 | | .163 | | .317 | |
| F | | 20.065*** | | 19.388** | | 21.039*** | | 68.842*** | |
| Dw | | 1.967 | | 1.974 | | 2.173 | | 2.001 | |
| Maximum VIF | | 1.035 | | 1.035 | | 1.252 | | 1.305 | |
| Minimum Tolerance | | .966 | | .966 | | .799 | | .766 | |

Note: * p < .05

** p < .01

*** p < .001.

of knowledge heterogeneity on innovation performance to some extent. The relationship conflict caused by knowledge heterogeneity will make individuals have subjective and emotional judgment when dealing with things. In organizational activities, both parties in the relationship conflict will ignore the objective judgment of the value of the idea in order to negate the idea holder under the subjective consciousness, which ultimately restricts the development of innovation performance.

Meanwhile, the dw values of the model are tested in Table 7, and the results are all within the standard range of 1.5 to 2.5, indicating that the samples in the scale are independent and there is no sequential correlation problem. The minimum tolerance in the model is greater than .5, and the maximum VIF value is less than 5, which proves that there is no collinearity between variables in the regression model and meets the requirements for model fitting.

*3. Parallel mediation test of bootstrapping.* Preacher & Hayes [60] has suggested that in order to obtain measurement results intuitively, multi-mediation models should be tested by Process SPSS. To further confirm the parallel mediating effect of task conflict and relationship conflict. The Bootstrapping test of knowledge heterogeneity, task conflict, relationship conflict and innovation performance were also conducted. The verification Process similarly takes into account four control variables, which are measured by Process v2.16.3. Model4 was adopted to conduct 5000 sampling times. The test results are shown in Table 8.

Through task conflict, the Indirect Effect of knowledge heterogeneity on innovation performance is significant (Indirect Effect = .107, SE = .037, BootLLCI = .038, BootULCI = .184).

Through relationship conflict, knowledge heterogeneity has a significant Indirect Effect on innovation performance (Indirect Effect = -.109, SE = .037, BootLLCI = -.191, BootULCI = -.043).

After controlling the mediating Effect, the Indirect effect of knowledge heterogeneity on innovation performance is still significant (Indirect Effect = .513, SE = .048, BootLLCI = .418, BootULCI = .608, t = 10.656, p < .001). It is verified that task conflict and relationship conflict

**Table 8.  Results of mediation effect analysis by bootstrapping (knowledge heterogeneity, task conflict, relationship conflict and innovation performance).**

|  |  | Effect | SE | t | p | BootLLCI | BootULCI |
|---|---|---|---|---|---|---|---|
| Direct Conflict |  | .513 | .048 | 10.656 | .000 | .418 | .608 |
| Indirect Conflict | TC | .107 | .037 |  |  | .038 | .184 |
|  | RC | -.109 | .037 |  |  | -.191 | -.043 |
| C1 (TC-RC) |  | .216 | .048 |  |  | .125 | .313 |

Note: 5000 times of sampling; Control variables have been added; Confidence interval 95%; The coefficients in the table are non-standardized coefficients.

play a partial mediating role on knowledge heterogeneity and innovation performance. On the one hand, knowledge heterogeneity in an organization may lead to viewpoint collision among members in work tasks, which helps the organization to optimize suggestions and promote innovation performance. But at the same time, it may also cause emotional conflicts between individuals outside of work, namely relationship conflicts. The generation of relationship conflict will destroy the harmonious communication atmosphere in the organization, inhibit the effective transfer of knowledge between organizations, and restrict the development of innovation performance.

**4.3.3 Test of moderated mediator model.**   This study intends to verify the moderating effect of knowledge sharing of moderating variables on the first half of the hypothesis Model (Fig 2.). Model 8 in Process v2.16.3 is used to test the moderated mediating effect, and the results are presented in Table 9.

**Table 9.  Moderated mediating effect analysis results.**

| Type | Variable | TC | | RC | | IP | |
|---|---|---|---|---|---|---|---|
|  |  | M1 | | M2 | | M3 | |
|  |  | B | t | B | t | B | t |
| Control Variable | Year | -.022 | -.542 | -.092 | -1.663 | .036 | 1.046 |
|  | Edu | .074 | .820 | -.173 | -1.424 | .244 | 3.261** |
|  | Age | -.017 | -.315 | -.576 | -7.909*** | -.077 | -1.566 |
|  | R&D | .003 | .066 | -.127 | -2.265* | -.003 | -.082 |
| Independent Variable | KH | .549 | 11.395*** | .224 | 3.464*** | .514 | 10.653*** |
| Mediator | TC |  |  |  |  | .167 | 3.589*** |
|  | RC |  |  |  |  | -.453 | -13.042*** |
| Moderator | KS | .092 | 2.087* | -.094 | -1.591 | .133 | 3.658*** |
| Int | KH*KS | .104 | 3.268** | -.111 | -2.598* | .079 | 2.960** |
| $R^2$ |  | .323 | | .251 | | .610 | |
| Adjust $R^2$ |  | .308 | | .235 | | .598 | |
| $\Delta R^2$ |  | .023 | | .016 | | .011 | |
| F |  | 21.509*** | | 15.146*** | | 54.473*** | |
| Dw |  | 1.967 | | 2.025 | | 2.007 | |
| Maximum VIF |  | 1.172 | | 1.172 | | 1.625 | |
| Minimum Tolerance |  | .853 | | .853 | | .615 | |

Note: * p < .05

** p < .01

*** p < .001; All variables in the model adopt the variables after centralization.

In M3, innovation performance is taken as the dependent variable and knowledge heterogeneity as the independent variable to explore the moderating effect of knowledge sharing on knowledge heterogeneity and innovation performance. As can be seen from the table, the product term of knowledge heterogeneity and knowledge sharing (B = .079, t = 2.960, p < .01) has a significant positive effect on innovation performance of the dependent variable. This means that knowledge sharing has a positive moderating effect between knowledge heterogeneity and innovation performance, and hypothesis 8 is valid. When the degree of knowledge sharing is higher, the knowledge interaction within the organization is intensified. In the rapid flow of knowledge, innovative ideas and high-quality decisions are more likely to emerge. Therefore, the stronger the influence of knowledge heterogeneity on innovation performance is. On the contrary, when the degree of knowledge sharing is low, knowledge heterogeneity cannot be transmitted in communication, and the positive impact on innovation performance is weak.

In M1, task conflict was taken as the dependent variable and knowledge heterogeneity as the independent variable to explore the moderating effect of knowledge sharing on knowledge heterogeneity and task conflict. As can be seen from the table, the product term of knowledge heterogeneity and knowledge sharing (B = .104, t = 3.268, p < .01) has a significant positive effect on dependent variable task conflict. This means that knowledge sharing has a significant positive moderating effect between knowledge heterogeneity and task conflict, and hypothesis 9 is valid. The higher the degree of knowledge sharing, the stronger the impact of knowledge heterogeneity on task conflict, the stronger the impact of knowledge heterogeneity on task conflict. The lower the degree of knowledge sharing, the weaker the influence of knowledge heterogeneity on task conflict.

In M2, with relationship conflict as the dependent variable and knowledge heterogeneity as the independent variable, the moderating effect of knowledge sharing on knowledge heterogeneity and relationship conflict was discussed. It can be seen from the table that the product term of knowledge heterogeneity and knowledge sharing (B = -.111, t = -2.598, p < .05) has a significant negative effect on dependent variable relationship conflict. Hypothesis 10 is true. When the degree of knowledge sharing is higher, the knowledge heterogeneity can be transmitted more harmoniously in the organization, the communication between members is more harmonious, and the relationship conflict is weaker. On the contrary, when the degree of knowledge sharing is lower, the heterogeneity of knowledge is more likely to cause emotional contradictions among members, which has a stronger impact on relationship conflicts.

In M3, the moderating effect of knowledge sharing on knowledge heterogeneity and innovation performance is discussed, taking innovation performance as the dependent variable and knowledge heterogeneity as the independent variable. As can be seen from the table, the product terms of knowledge heterogeneity and knowledge sharing (B = .079, t = 2.960, p < .01) have a significant positive impact on innovation performance of the dependent variable. Let's say 8 is true. The higher the degree of knowledge sharing, the better the effect of knowledge heterogeneity on innovation performance. On the contrary, when the degree of knowledge sharing is low, the heterogeneity of knowledge is more likely to restrict the improvement of innovation performance due to poor communication.

**4.3.4 Slope analysis.** In order to further understand the relationship between adjustment effects, slope analysis is also carried out for the adjustment effects with significant effects. According to Process v2.16.3, the average value is ± 1 standard deviation, and the trend chart under different knowledge sharing environments is drawn, so as to more intuitively see the regulating effect of knowledge sharing on knowledge heterogeneity and task conflict, as well as on knowledge heterogeneity and innovation performance.

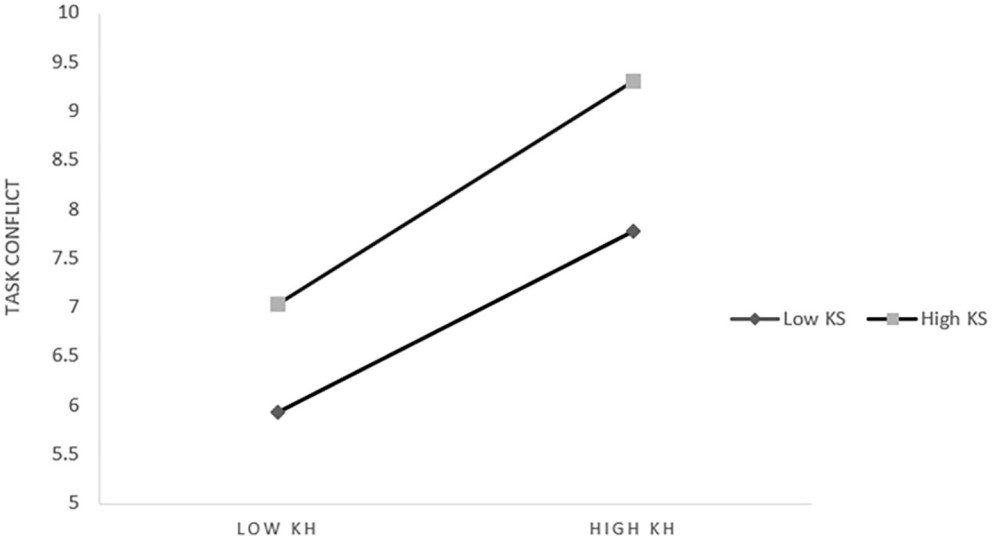

**Fig 3. The moderating effect of knowledge sharing on knowledge heterogeneity and task conflict.**

*1. The moderating effect of knowledge sharing on knowledge heterogeneity and task conflict.* Further slope analysis shows that Fig 3, from which we can see: When knowledge sharing is at a low level (M-1SD) (Simple slope = 0.436, t = 4.022, $p < 0.001$), knowledge heterogeneity has a significant positive impact on task conflict. When knowledge sharing is at a high level (M +1SD) (Simple slope = 0.662, t = 7.259, $p < 0.001$) Knowledge heterogeneity had a significant positive effect on task conflict, which was higher than that at the low level. It shows that with the improvement of knowledge sharing level, the influence of knowledge heterogeneity on task conflict shows an increasing trend, which indicates that it has a positive moderating effect.

*2. The moderating effect of knowledge sharing on knowledge heterogeneity and relationship conflict.* Further slope analysis shows that Fig 4 is shown, from which we can see: When knowledge sharing is at a low level (M-1SD) (Simple slope = 0.345, t = 4.157, $p < 0.001$), knowledge

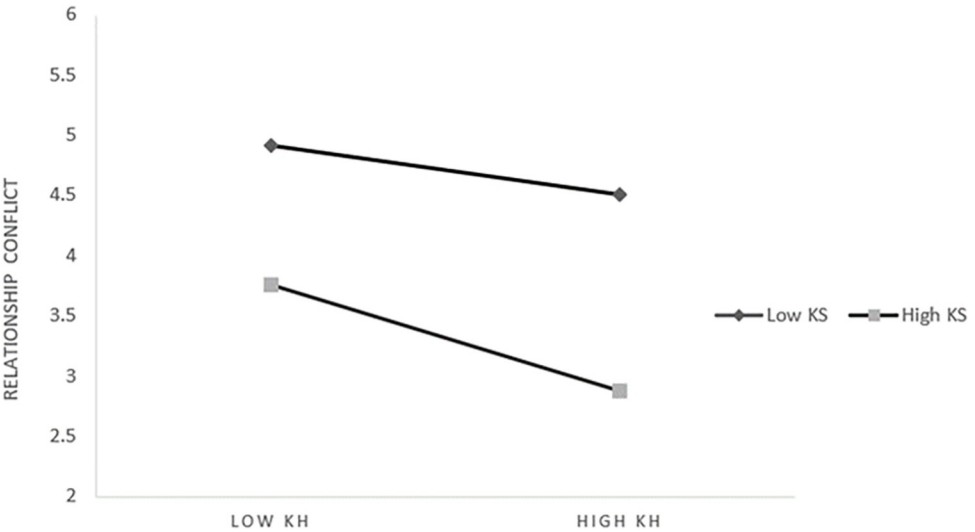

**Fig 4. The moderating effect of knowledge sharing on knowledge heterogeneity and relationship conflict.**

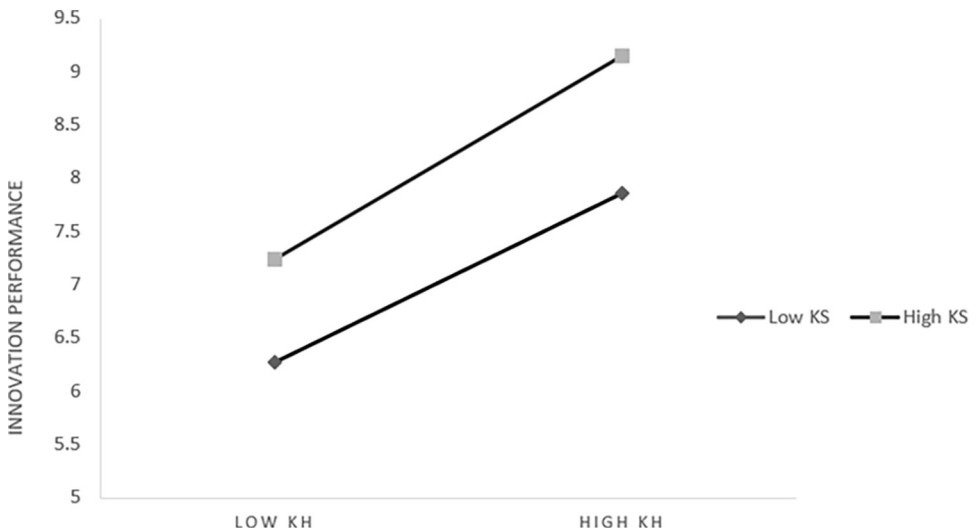

**Fig 5. The moderating effect of knowledge sharing on knowledge heterogeneity and performance innovation.**

heterogeneity has a significant positive impact on relationship conflict. When knowledge sharing is at a high level (M+1SD) (Simple slope = 0.103, t = 0.925, $p > 0.05$) Knowledge heterogeneity had no significant effect on relationship conflict. Only at a low level of knowledge sharing can the influence of knowledge heterogeneity on the relationship be moderated effectively. In other words, the lack of effective knowledge transfer within the organization will further aggravate the interpersonal conflicts caused by knowledge differences. But it also doesn't mean that, as long as there is sufficient sharing, no amount of knowledge difference can cause emotional conflict.

*3. The moderating effect of knowledge sharing on knowledge heterogeneity and innovation performance*. Further slope analysis shows that Fig 5, from which we can see: When knowledge sharing is at a low level (M-1SD) (Simple slope = 0.428, t = 4.615, $p < 0.001$), knowledge heterogeneity has a significant positive impact on innovation performance. When knowledge sharing is at a high level (M+1SD) (Simple slope = 0.601, t = 6.468, $p < 0.001$) Knowledge heterogeneity had a significant positive effect on innovation performance, which was higher than that at the low level. It shows that with the improvement of knowledge sharing level, the influence of knowledge heterogeneity on innovation performance shows an increasing trend, which indicates that it has a positive moderating effect.

## 5. Discussion

The final results (Table 10.) well explain the three research questions raised in this study. For the RQ1, although some research suggests that knowledge heterogeneity has an inverted U-shaped influence on innovation performance. Taking into account the Chinese context, this study posits that China, a diverse country, particularly the region of Guizhou—where various ethnicities coexist—naturally has a high level of cultural, knowledge, and ideological heterogeneity. Enterprises operating within this environment over time have likely developed a strong capacity for accepting and tolerating heterogeneous knowledge. Hence, concerns such as organizational disarray due to excessive heterogeneity are less likely. The results of hypothesis 1 also support this conclusion. The introduction of heterogeneous knowledge can assist the region, particularly SMEs, in gaining broader knowledge and information. In a rapidly changing information landscape, enterprises can also timely grasp current market dynamics and

Table 10. Research results.

| Hypothesis | Content | Result |
|:---:|---|:---:|
| H1 | There is a positive correlation between knowledge heterogeneity and firm innovation performance. | Valid |
| H2 | Knowledge heterogeneity positively influences task conflict. | Valid |
| H3 | Task conflict positively impacts firm innovation performance. | Valid |
| H4 | Task conflict mediates the positive relationship between knowledge heterogeneity and firm innovation performance. | Valid |
| H5 | Knowledge heterogeneity positively affects relationship conflict. | Valid |
| H6 | Relationship conflict negatively impacts firm innovation performance | Valid |
| H7 | Relationship conflict mediates the relationship between knowledge heterogeneity and firm innovation performance, albeit negatively. | Valid |
| H8 | Knowledge sharing strengthens the positive relationship between knowledge heterogeneity and innovation performance. | Valid |
| H9 | Knowledge sharing augments the positive relationship between knowledge heterogeneity and task conflict. | Valid |
| H10 | Knowledge sharing diminishes the positive relationship between knowledge heterogeneity and relationship conflict. | Valid |

develop a keen ability to forecast future market trends. Innovation performance is primarily reflected in the development of new products, services, and markets.

In order to explain RQ2, this study attempts to explore a series of internal organizational processes before the heterogeneous knowledge transforming into innovation performance, based on the Input-Process-Output (IPO) model logic. Factors such as internal members, managers, or even external environment could influence the final innovation performance during this transformation. It is thus essential to consider these organizational conflicts from a two-sided perspective.

Knowledge heterogeneity, involving differences in thought, cognition, and ideology within a group, can lead to divergent approaches to work tasks, resulting in task conflict. Through the exchange of ideas arising from knowledge heterogeneity, organizational members can reflect, overcome limited thinking, explore new perspectives, and ultimately promote innovation performance. The data, as supported by M3 and M4 in the Table 6, as well as Table 8, shows that task conflict plays a partial mediating role, validating this inference.

Relationship conflict, in contrast to task conflict, is more subjective in nature. On one hand, during organizational communication and discussion related to enterprise innovation, interpersonal communication frictions can lead members to subjectively reject some suggestions and opinions put forward by others, thereby restricting enterprise innovation and development. On the other hand, the presence of relationship conflicts in the daily workflow can create a rigid organizational atmosphere and reduce cooperation efficiency among members. Even when breakthrough ideas emerge in an organization, inadequate execution due to these conflicts can negatively impact innovation performance. This is demonstrated in M3 and M4 in Table 7, as well as Table 8, which show that relationship conflict has a negative effect on firm innovation and plays a partial mediating role in the relationship between knowledge heterogeneity and firm innovation performance. This implies that even with high degrees of knowledge heterogeneity, strong relationship conflict can limit the ultimate performance of organizational innovation.

For the RQ3, this study also examines knowledge sharing as an organizational state and explores whether a good sharing atmosphere can enhance the impact of knowledge heterogeneity on innovation performance within organizations. It confirms that a positive knowledge

sharing atmosphere can effectively moderate the influence of knowledge heterogeneity on task conflict, relationship conflict, and innovation performance. In an organization with a healthy knowledge sharing state, knowledge heterogeneity can quicken the iteration of information and knowledge through efficient sharing behavior, and directly regulate innovation performance. Simultaneously, it can encourage members to focus more on task discussion and actively provide work suggestions, promoting the emergence of task conflicts.

However, although knowledge sharing has shown to moderate the relationship between knowledge heterogeneity and task conflict, regression analysis results indicate that high-level knowledge sharing does not significantly moderate the relationship between knowledge heterogeneity and relationship conflict. That is, a positive knowledge sharing atmosphere does not necessarily mitigate emotional contradictions sparked by individual heterogeneity. This could be due to the strong collectivist atmosphere prevalent in Chinese enterprises, where a good sharing environment may only maintain superficial organizational harmony, without necessarily facilitating genuine agreement among employees. As these disagreements accumulate, relationship conflict could still erupt.

These results highlight a need for future research to pay closer attention to the concealed behaviors and personal feelings of individuals when considering interpersonal issues within the Chinese context. Such concealment can create misleading appearances, which could disrupt accurate research findings.

## 6. Implication and limitation

Firstly, within the framework of the IPO model and resource-based theory, this study analyzes the impact mechanism of knowledge heterogeneity on innovation performance. It introduces task conflict and relationship conflict as parallel mediators, focusing on the impact of the intervention of heterogeneous knowledge on the existing organizational order and shared mental models. At the same time, it considers knowledge sharing as a moderating variable to expand the study of situational factors under the model mechanism.

From a practical perspective, this study focuses on the innovation development issues of small and medium-sized manufacturing enterprises in Guizhou region, effectively combining local realities with theoretical knowledge. The research data points out that compared to relying on external funds and technological input, such enterprises should fully exploit the local multi-ethnic and multicultural characteristics to carve out their unique development path.

However, this study still has certain limitations. Firstly, the mediating variables of this study have not fully explained the inherent mechanism of knowledge heterogeneity on innovation performance. Future studies could expand the research of mediating variables to further perfect the theoretical model. Secondly, this study only discusses the moderating role of knowledge sharing. Future research can consider enriching the moderating variables. Thirdly, the subjects of this study are primarily focused on small and medium-sized manufacturing enterprises in Guizhou province, which has certain regional limitations. Future research could add cross-industry research and long-term enterprise tracking research based on this, which would validate the theoretical model more effectively.

## Author Contributions

**Data curation:** Xiaoheng Chen.

**Formal analysis:** Xiaoheng Chen.

**Investigation:** Rui Gan, Zhiyan Wang.

**Methodology:** Xiaoheng Chen.

**Resources:** Zhiyan Wang.

**Software:** Rui Gan.

**Supervision:** Xing Zhang.

**Writing – review & editing:** Xing Zhang.

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
