## [Decision Letter · Decision Letter 0]

28 Jun 2023

PONE-D-23-14521Heterogeneity of Knowledge and Corporate Innovation Performance: The Mediating Influence of Task Conflict and Relationship ConflictPLOS ONE

Dear Dr. ZHANG,

Thank you for submitting your manuscript to PLOS ONE. After careful consideration, we feel that it has merit but does not fully meet PLOS ONE’s publication criteria as it currently stands. Therefore, we invite you to submit a revised version of the manuscript that addresses the points raised during the review process.

We look forward to receiving your revised manuscript.

Kind regards,

Rana Muhammad Ammar Zahid, PhD

Academic Editor

PLOS ONE

4. Please include a copy of Tables 8, 13, 22 and 23 which you refer to in your text on pages 9,17 and 22.

Additional Editor Comments:

Dear Authors,

There are some serious issues regarding the research gap, hypothesis developments, discussions of results. I agreed with the reviewers and invite you to submit the revised draft after incorporating the changes recommended by the reviewers.

Reviewers' comments:

Reviewer's Responses to Questions

**Comments to the Author**

1. Is the manuscript technically sound, and do the data support the conclusions?

Reviewer #1: Yes

Reviewer #2: Yes

2. Has the statistical analysis been performed appropriately and rigorously? 

Reviewer #1: Yes

Reviewer #2: Yes

3. Have the authors made all data underlying the findings in their manuscript fully available?

Reviewer #1: Yes

Reviewer #2: Yes

4. Is the manuscript presented in an intelligible fashion and written in standard English?

Reviewer #1: Yes

Reviewer #2: Yes

5. Review Comments to the Author

Reviewer #1: 1. Compared with the existing achievements in the established research field, the uncertainty and exploratory nature of the hypotheses put forward in the manuscript are low, and the progress of the research results is not obvious.

2. In the manuscript, before the authors put forward the questions to be studied, the groundwork is insufficient, especially in terms of content pertinence and logical completeness.

3. The content of the Literature Review part of the manuscript is too basic and scattered, and the authors failed to focus on the specific issues to be studiedand carry out targeted, systematic and in-depth literature research.

4. In the empirical research section, the manuscript does not pay enough attention to the problems, causes and targeted improvements in the corresponding areas of practice. This limits the application value of this article.

5. In addition, the title of the manuscript does not correspond completely to its content.

Reviewer #2: • Recommendation: Major Revision

Dear author/s,

Thank you for the opportunity to review this manuscript. I hope you find my comments useful as you consider revising the paper. The topic is fitting with the aim and scope of the Journal. I hope this review provides some useful feedback and wish you the best of luck with the development of this paper!

Additional Questions:

1. Originality: Does the paper contain new and significant information adequate to justify publication?: Yes. he authors identified an interesting topic and provided significant information on the relation of "Heterogeneity of Knowledge and Corporate Innovation Performance: The Mediating Influence of Task Conflict and Relationship Conflict”.

However, there are some issues that need clarification. What is the motivation of this paper? In the introduction I do not understand and see clearly the contribution of the paper. I think the paper, at the present form, partially fails to formulate a research problem, which is of interest.

What is the research gap? We have partial answers on what we know now about the topic and what we do not know. The author should more in detail and in a more systematic way present answer on these questions, but also what we need to know. Why is this important, for research, for practice? Implication part is missing.

Introduction should be upgraded. First, I suggest supporting more key sentences with proper literature (and to clearly theoretically positioning your paper). Second, connect more the key paragraphs (a clear red thread should be created). Third, the introduction is critical and I suggest the following key points within this section (Positioning, Gap, Purpose, Central argument, Organizing, Contribution, So what?).

You also need to argue a brief summary of methodology and findings/contributions, each in one paragraph.

2. Relationship to Literature: Does the paper demonstrate an adequate understanding of the relevant literature in the field and cite an appropriate range of literature sources? Is any significant work ignored?: The literature review is not fine. Kindly update literature by adding new studies. Why you conduct this research? Properly explain in this section. Theory is a most important part of article. I don’t see that authors explain regarding theory. Its my suggestions to add separate section of theory in literature and connect theory with research framework.

3. Methodology: Is the paper's argument built on an appropriate base of theory, concepts, or other ideas? Has the research or equivalent intellectual work on which the paper is based been well designed? Are the methods employed appropriate?: Yes

4. Results: Are results presented clearly and analysed appropriately? Do the conclusions adequately tie together the other elements of the paper?: The results are very descriptive and contain more statistical explanation than clear application.

5. Discussion, Implications for research, practice and/or society: Does the paper identify clearly any implications for research, practice and/or society? Does the paper bridge the gap between theory and practice? How can the research be used in practice (economic and commercial impact), in teaching, to influence public policy, in research (contributing to the body of knowledge)? What is the impact upon society (influencing public attitudes, affecting quality of life)? Are these implications consistent with the findings and conclusions of the paper?:

I would suggest that a discussion section be more comprehensively developed that links back to your initial research questions and a clear statement of proposed contributions, once you have reframed your arguments and developed some propositions. What should we, as readers, take away regarding your study? What are the key theoretical contributions that are gained? How can these findings contribute to the literature stream associated with green behavior? What do we know about this literature stream now that we have read your study? What future research should be conducted within this literature stream that can be extended based upon your study?

This is what I often call "closing the loop". Specifically, you a) state in the introduction that there is a gap (your research questions), and you plan to address the gap theoretically; b) present a formally developed and very focused literature review that gives the rational for the study and develop propositions/hypos that reflect this gap; and c) "Close the loop", by developing your discussion section that ties back to the research question(s). In the end, you hope that the reader has been able to read the article and see the article, in its entirety, as encapsulating the resolution of a theoretical or empirical gap.

6. Quality of Communication: Does the paper clearly express its case, measured against the technical language of the field and the expected knowledge of the journal's readership? Has attention been paid to the clarity of expression and readability, such as sentence structure, jargon use, acronyms, etc.: A professional review of the language is strongly suggested because several parts of the text are unclear.

6. PLOS authors have the option to publish the peer review history of their article (what does this mean?). If published, this will include your full peer review and any attached files.

Reviewer #1: No

Reviewer #2: No

---

## [Author Response · Author response to Decision Letter 0]

9 Aug 2023

Dear editors and reviewers:

Thank you for your letter and the reviewers’ comments on my manuscript entitled " Heterogeneity of Knowledge and Corporate Innovation Performance: The Mediating Influence of Task Conflict and Relationship Conflict" (ID: PONE-D-23-14521). Those comments are very helpful for revising and improving our paper, as well as the important guiding significance to our future research. We have studied the comments carefully and made corrections which we hope meet with approval. The responds to the reviewers’ comments are as follows.

Replies to the reviewers’ comments:

Reviewer #1:

Thank you very much for your constructive comments and suggestions which would help us in depth to improve the quality of the paper.

1.“ Compared with the existing achievements in the established research field, the uncertainty and exploratory nature of the hypotheses put forward in the manuscript are low, and the progress of the research results is not obvious.”

Response: Thank you for your advice. Your feedback helps us better articulate the value of the hypotheses proposed in our study and present our findings more effectively to readers. Therefore, we have made two changes. First, to enhance the uncertainty and exploratory nature of our research hypotheses, we added discussions on their theoretical significance in the Introduction section. We also emphasized the uncertainty of the hypotheses and the necessity to further explore and discuss in the context of China in the Literature Review. Second, we have added Table 11 titled "Research Results" to summarize our findings.

2. “In the manuscript, before the authors put forward the questions to be studied, the groundwork is insufficient, especially in terms of content pertinence and logical completeness.”

Response: Thank you for your advice. To better showcase the foundational work of our research, we have made two additions: 1. We have included a literature review of the resource-based theory and clarified its implications for our study. 2. We further specified the research questions in the Introduction section.

3. “The content of the Literature Review part of the manuscript is too basic and scattered, and the authors failed to focus on the specific issues to be studied and carry out targeted, systematic and in-depth literature research.”

Response: Thank you for your advice. To better demonstrate the specificity, comprehensiveness, and depth of our literature research, we have made significant revisions to the literature review section.

4. “In the empirical research section, the manuscript does not pay enough attention to the problems, causes and targeted improvements in the corresponding areas of practice. This limits the application value of this article.”

Response: Thank you for your advice. In order to better address the application value of this article, we made modifications in two sections. First, in the Introduction, we added discussions on the theoretical and practical significance. Second, we added a separate chapter titled "Implications and Limitations" to explain the overall significance and limitations of our research.

5. “In addition, the title of the manuscript does not correspond completely to its content.”

Response: Thank you for your advice. To better demonstrate the relevance between our topic and research content, we made a slight adjustment to the expression of the paper title. We changed the title to: "Knowledge Heterogeneity and Corporate Innovation Performance: The Mediating Influence of Task Conflict and Relationship Conflict."

Reviewer #2:

I really appreciate your recognition and encouragement to my study. Thank you very much for your constructive comments and suggestions which would help us in depth to improve the quality of the paper.

1.“What is the motivation of this paper? In the introduction I do not understand and see clearly the contribution of the paper. I think the paper, at the present form, partially fails to formulate a research problem, which is of interest. What is the research gap? We have partial answers on what we know now about the topic and what we do not know. The author should more in detail and in a more systematic way present answer on these questions, but also what we need to know. Why is this important, for research, for practice? Implication part is missing. Introduction should be upgraded. First, I suggest supporting more key sentences with proper literature (and to clearly theoretically positioning your paper). Second, connect more the key paragraphs (a clear red thread should be created). Third, the introduction is critical and I suggest the following key points within this section (Positioning, Gap, Purpose, Central argument, Organizing, Contribution, So what?).

You also need to argue a brief summary of methodology and findings/contributions, each in one paragraph.”

Response: Thank you for your advice. Based on your suggestions, we made significant revisions to the Introduction section. We made three main modifications: 1. We reduced unnecessary explanations in the Introduction. 2. We identified the research gap. 3. We added discussions on the theoretical and practical significance, clarifying the research questions.

2.“Does the paper demonstrate an adequate understanding of the relevant literature in the field and cite an appropriate range of literature sources? Is any significant work ignored ?: The literature review is not fine. Kindly update literature by adding new studies. Why you conduct this research? Properly explain in this section. Theory is a most important part of article. I don’t see that authors explain regarding theory. Its my suggestions to add separate section of theory in literature and connect theory with research framework.”

Response: Thank you for your advice. Based on your suggestions, we made significant revisions to the Literature Review section. Specifically, addressing your feedback, we added a literature review of the resource-based theory and clarified its implications for our study.

3.“Discussion, Implications for research, practice and/or society: Does the paper identify clearly any implications for research, practice and/or society? Does the paper bridge the gap between theory and practice? How can the research be used in practice (economic and commercial impact), in teaching, to influence public policy, in research (contributing to the body of knowledge)? What is the impact upon society (influencing public attitudes, affecting quality of life)? Are these implications consistent with the findings and conclusions of the paper? I would suggest that a discussion section be more comprehensively developed that links back to your initial research questions and a clear statement of proposed contributions, once you have reframed your arguments and developed some propositions. What should we, as readers, take away regarding your study? What are the key theoretical contributions that are gained? How can these findings contribute to the literature stream associated with green behavior? What do we know about this literature stream now that we have read your study? What future research should be conducted within this literature stream that can be extended based upon your study?”

Response: Thank you for your advice. Based on your suggestions, we made significant revisions to the Discussion section. We made three main modifications: 1. We further clarified the research questions and echoed the Research Questions at the end. 2. We added an "Implication and Limitation" section, making the research logic more complete, forming a "closing the loop" structure. 3. In both the Introduction and Implication sections, we added discussions about the practical significance and exploration of the research gap.

4.“ Quality of Communication: Does the paper clearly express its case, measured against the technical language of the field and the expected knowledge of the journal's readership? Has attention been paid to the clarity of expression and readability, such as sentence structure, jargon use, acronyms, etc.: A professional review of the language is strongly suggested because several parts of the text are unclear.”

Response: Thank you for your advice. Based on your suggestions, we further examined the accuracy and standardization of the language used in the paper. We polished the overall language to make the article more in line with academic requirements, enhancing its readability.

Once again, thank you very much for your constructive comments and suggestions which would help me in depth to improve the quality of the paper. We have studied the comments carefully and made corrections which we hope meet with approval.

Your sincerely,

Xing Zhang

School of Management, Guizhou University of Commerce, China

Email: 201710616@gzcc.edu.cn

---

## [Decision Letter · Decision Letter 1]

18 Sep 2023

Knowledge Heterogeneity and Corporate Innovation Performance: The Mediating Influence of Task Conflict and Relationship Conflict

PONE-D-23-14521R1

Dear Dr. ZHANG,

We’re pleased to inform you that your manuscript has been judged scientifically suitable for publication and will be formally accepted for publication once it meets all outstanding technical requirements.

Kind regards,

Rana Muhammad Ammar Zahid, PhD

Academic Editor

PLOS ONE

Additional Editor Comments (optional):

Reviewers' comments:

Reviewer's Responses to Questions

**Comments to the Author**

1. If the authors have adequately addressed your comments raised in a previous round of review and you feel that this manuscript is now acceptable for publication, you may indicate that here to bypass the “Comments to the Author” section, enter your conflict of interest statement in the “Confidential to Editor” section, and submit your "Accept" recommendation.

Reviewer #1: All comments have been addressed

Reviewer #2: All comments have been addressed

2. Is the manuscript technically sound, and do the data support the conclusions?

Reviewer #1: Yes

Reviewer #2: Yes

3. Has the statistical analysis been performed appropriately and rigorously? 

Reviewer #1: Yes

Reviewer #2: Yes

4. Have the authors made all data underlying the findings in their manuscript fully available?

Reviewer #1: Yes

Reviewer #2: Yes

5. Is the manuscript presented in an intelligible fashion and written in standard English?

Reviewer #1: Yes

Reviewer #2: Yes

6. Review Comments to the Author

Reviewer #1: The comments or suggestions from the previous round of reviews on this article have been responded to in a targeted manner.

Reviewer #2: Dear authors: I hope you are doing well. Your article is now improved.

No further comments from my side. Good luck

7. PLOS authors have the option to publish the peer review history of their article (what does this mean?). If published, this will include your full peer review and any attached files.

Reviewer #1: No

Reviewer #2: No

---

## [Editor Report · Acceptance letter]

25 Sep 2023

PONE-D-23-14521R1 

Knowledge Heterogeneity and Corporate Innovation Performance: The Mediating Influence of Task Conflict and Relationship Conflict 

Dear Dr. Zhang:

I'm pleased to inform you that your manuscript has been deemed suitable for publication in PLOS ONE. Congratulations! Your manuscript is now with our production department. 

Kind regards, 

on behalf of

Dr. Rana Muhammad Ammar Zahid 

Academic Editor

PLOS ONE